# Aesthetic and Thermal Suitability of Highly Glazed Spaces with Interior Roller Blinds in Najran University Buildings, Saudi Arabia

Abdultawab M. Qahtan

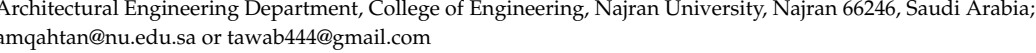

Architectural Engineering Department, College of Engineering, Najran University, Najran 66246, Saudi Arabia;
amqahtan@nu.edu.sa or tawab444@gmail.com

**Abstract:** Highly glazed spaces are visually appealing and trendy, but effectively managing their temperature in hot arid climates remains a significant challenge. This study evaluates the effectiveness of dark-tinted double low-E glass with internal roller blinds in reducing heat gain in glazed spaces in hot arid climates and investigates architects' perspectives on these facades. It combines field measurements and a survey to assess the balance between thermal control and aesthetics in such environments. This study reveals that the current glazing significantly attenuates solar radiation ingress, evidenced by a marked indoor—outdoor temperature differential ($\Delta T$) of approximately 9.2 °C. The mean radiant temperature registers at 1.5 °C above the indoor air temperature, which can be attributed to the glazing's propensity to absorb and retain solar heat, resulting in an inner glass surface temperature of 43 °C. The implementation of adjustable blinds has a dynamic influence on the heat transfer coefficient (HTC), effectively modulating the temperature by impeding natural convection currents. With the blinds retracted, the HTC stands at an average of 7.1 W/m$^2$K, which diminishes to 5 W/m$^2$K when the blinds are 50% closed and further reduces to 4.2 W/m$^2$K when the blinds are fully closed (100%). Survey results suggest that architects prioritise glazed facades for aesthetics (52%) while facing challenges in thermal and energy efficiency (44%). Future studies should concentrate on developing novel glazing systems that integrate solutions for visual appeal, lighting and thermal efficiency in glazed facades, particularly in hot arid climates.

**Keywords:** tinted glazing double low-E; interior blinds; thermal profile; hot arid climate; aesthetic appeal

## 1. Introduction

From the standpoint of architectural design, the facade stands out as a crucial element in a building's ability to display its aesthetic qualities and convey its architectural identity. Meanwhile, from an engineering standpoint, building envelopes, which encompass the facade, assume a vital role in preserving indoor thermal profiles and enhancing the overall sustainability of buildings [1]. Among the critical aspects of architecture, building envelope materials play a pivotal role in shielding interior spaces from the harsh effects of outdoor environments, particularly in the context of excessive heat gain. The evolution of building facades over history is a testament to their adaptability to meet functional and climatic demands. From their humble origins, using materials such as clay, stone, wood and brick, facades have progressed to incorporate steel and glass, reflecting advancements in technology and design [2]. These changes in materials are driven not only by practical considerations but also by the pursuit of architectural beauty and aesthetics. In the thoughtful choice of materials for building facades, a crucial task is to recognise that the visual appeal of these materials holds a significance that goes beyond mere comfort [3]. Notably, exterior aesthetics often wield a more pronounced influence than their interior counterparts, captivating attention and shaping perceptions [4]. On the basis of this concept, the aesthetics of extensively glazed facades have been adopted worldwide. However, since the early days of modern architecture, concerns have been raised about their associated drawbacks,

including issues such as glare, increased thermal loads and the need for external shading solutions [5]. The following review briefly outlines the aspects previously discussed in research on the factors that influence an architect's choice of a glazed facade, even when it contradicts the primary design recommendations.

### 1.1. Aesthetics Appeal of Glazed Facades

The ability of architectural forms to evoke a deep sense of beauty is often referred to as the 'aesthetic' function [6]. Glass, with its extensive aesthetic possibilities, has become an integral part of contemporary building facades, empowering architects' creativity [7]. This infusion of glass into architectural design highlights the convergence of form and aesthetics, where the inherent beauty of glass blends seamlessly with the architect's vision, resulting in buildings that are both visually captivating and functionally sound. This situation aligns with the enduring principles of firmitas (strength), utilitas (utility) and venustas (beauty) established by the Roman architect Vitruvius in his influential work 'De architectura' [8]. These principles continue to guide architectural considerations through the ages. Aesthetic qualities, defined as those that evoke delight and admiration, significantly contribute to an object's appeal, making it visually attractive and beautiful [4]. Buildings with extensive glazing create a visual connection between the interior and exterior while simultaneously shielding occupants from outdoor weather conditions. This aspect holds great importance for architects, who often place great emphasis on the symbolic connotations associated with different design choices [9]. An iconic historical example of a completely glazed facade is the Crystal Palace, a creation of the architect Sir Joseph Paxton, constructed in London for the Great Exhibition of 1851 [10]. The prevalence of extensively glazed buildings has become a worldwide design trend, surpassing concerns related to climate [11]. Given this trend, engaging with the ongoing scholarly discourse about the influence of glazed facades on building design, encompassing aesthetics and beyond, becomes crucial.

When contemplating facade design in hot climates, three key factors come into play: aesthetics, thermal function and their impact on a building's energy consumption [1]. Numerous studies have delved into this subject, underscoring the complexity of glazed facades within architectural design. One study underscored the need to broaden our comprehension beyond comfort and energy efficiency, urging the inclusion of an aesthetic dimension in facade and space design [12]. A study centred on the morphology of glazed facades, with a specific focus on their aesthetic aspects, determined that the glazed facade and shading are two integral components that need to harmonise seamlessly [13]. The aesthetic elements are frequently neglected in the design of energy-efficient buildings, leading to visually unappealing outcomes on numerous occasions. The recognition of the role of glazing in shaping building aesthetics and its simultaneous influence on energy consumption underscores the dual importance of employing this material in building facades [4,9,14,15]. Architects often face trade-offs between aesthetics and heat gain, particularly in hot climates. While extensive glazing may provide stunning views and aesthetics, it can also result in higher thermal loads [5]. A comprehensive review of different glazing solutions also contributes to this discourse, assessing them from multiple perspectives, including their environmental impact and aesthetic influence [16]. These studies collectively illustrate that while glazed facades are often appreciated for their visual appeal, the specific effects of various types of glazing on building aesthetics remain a subject of intricate research and debate in the context of architecture and building design, particularly in hot arid climates like that of Najran University.

The architectural exteriors of Najran University buildings, as illustrated in Figure 1, exhibit a noticeable dichotomy: opaque facades are primarily constructed using prefabricated concrete, while glazed facades predominantly feature double layers of blue–green solar-control glass materials. The choice between highly glazed facades and a multitude of smaller windows plays a significant role in shaping a building's aesthetics. However, the primary concern revolves around evaluating the thermal efficiency of the highly glazed spaces, especially in the challenging context of a hot arid climate. As one walks through

the university campus buildings, a detail that becomes evident is that areas with extensive glazing are present in various locations, such as lecture halls, the library reading hall, lounges, corridors and administrative offices.

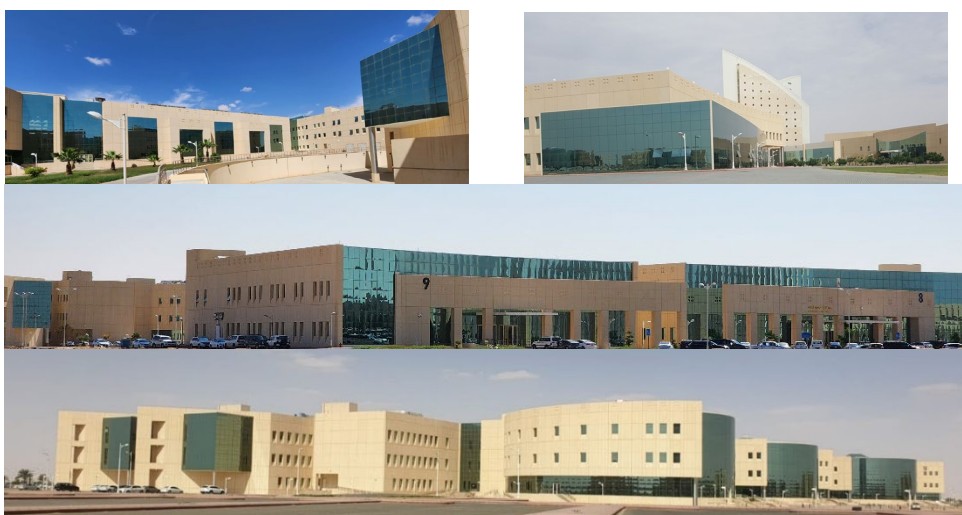

**Figure 1.** Exterior views of certain Najran University buildings highlighting glazed facades.

Generally, highly glazed facades not only offer a stylish and contemporary look but also allow ample natural light indoors, decreasing the need for artificial lighting during daylight hours. Moreover, they provide extensive views, creating a feeling of connectivity between indoor and outdoor areas, which collectively enhances the overall sense of well-being. The challenges associated with the use of aesthetically pleasing, highly glazed spaces in hot arid climates stem from the fact that these materials amplify the heat load within adjacent interior spaces, necessitating a significant amount of energy to maintain comfort. This condition has a profound impact on the nation's energy consumption, with Saudi Arabia reporting that the building sector is accountable for about 29% of total energy usage and more than 75% of total electricity consumption [17,18]. In this context, architectural design emerges as a crucial factor. Thoughtful and efficient design is essential for boosting energy efficiency and promoting sustainability in Saudi Arabian buildings. This study focuses on evaluating the thermal performance of highly glazed spaces at Najran University and exploring architects' viewpoints on the use of glazing. It serves as a case study, providing insights into regional architectural practices and their alignment with energy-efficient and sustainable design principles.

### 1.2. Solar Heat Gain through Glazing with Interior Roller Blinds

Solar heat gain is a major concern in hot arid climates, as it can lead to high indoor temperatures and increased energy consumption. Solar heat gain can be reduced in a number of ways, such as using advanced solar control glazing and installing shading devices. The amount of total solar radiation that passes through glass can be characterised in two ways. Firstly, it accounts for the heat gain caused by direct solar radiation transmission (ultraviolet [UV], visible light [VL] and infrared [IR]), represented as $\tau_s$, and secondly, it considers the inward heat transfer, denoted as Ni, which emerges due to the temperature disparity between the air temperature and the surfaces of the glazing. The total solar heat gain that permeates through glazing is typically assessed using the solar heat gain coefficient (SHGC) defined by ASHRAE in the following Equation (1) [19]:

$$\text{SHGC} = \tau_s + \text{Ni} \times \alpha s \qquad (1)$$

where $\tau_s$ is the direct solar transmittance of the fenestration system, Ni is the fraction of absorbed radiation that flows inward, and $\alpha s$ is the solar absorptance.

In hot climates, optimal glazing selection prioritises a low SHGC, and this choice is contingent upon the spectral attributes of the glass. Various glass types exhibit distinct characteristics in relation to solar radiation preferences for double low-emissivity (low-E), vacuum glazing and smart glass systems that autonomously adjust the transparency to effectively counteract heat gain [20]. Currently, low-E glass remains the most popular energy-efficient glass on the market. U-values in low-E glazing systems can be lowered through improvements made by adjusting gap thickness, altering inert gases, incorporating translucent aerogel materials and introducing multiple layers of low-E coating [21,22]. Low-E glass reflects heat and is evaluated by its U-value, with a lower U-value indicating better thermal insulation. The U-value measures heat transfer and is expressed as $W/m^2 \cdot K$. [7]. In hot climates, low-E coatings are commonly used on the outer pane to combat high outdoor temperatures during the summer, minimising UV damage, all with minimal impacts on natural light [23]. Observations indicate that buildings incorporating hard-coat low-E double glazing can achieve energy savings of approximately 9% to 14% in daily air-conditioner usage [24,25]. In brief, advancements in glazing are set to offer better choices for designing building facades, with a particular emphasis on minimising heat gain and enhancing visual comfort and the aesthetic appeal of facades [1].

The shortcomings of unshaded glass facades have long been recognised in the field of sustainable architecture for their impact on thermal discomfort [5]. Despite the use of advanced glazing, a certain amount of direct solar radiation can still infiltrate a building. Consequently, the strategic incorporation of shading devices is vital in achieving optimal control over solar heat gain. External shading is typically the most effective solar control method, but it may be less preferred due to factors such as cost, aesthetics, maintenance, structural constraints and wind load concerns [26]. The adoption of external window shading can conflict with architects' desires for transparent glass buildings [5]. However, in desert environments, interior blinds are a prudent option due to their immunity to external dust accumulation. Internal shading devices can affect the thermal performance of glazing by disrupting the natural convective airflow and reducing the exchange of long-wave radiation heat between the glazing and the indoor area. In addition to decreasing thermal radiation from the window surface, blinds act as a thermal barrier for individuals near the window [13,27–29]. Numerous studies have explored the advantages and considerations of utilising interior roller blinds as shading solutions in hot climates, as cited in References [30–33]. This study specifically examines the utilisation of top-down interior roller opaque blinds in spaces throughout Najran University.

As referenced in the review, the widespread adoption of glazing in building facades, driven by aesthetic and other considerations, has posed challenges in preserving optimal thermal comfort within buildings. In response, this study seeks to address the following research questions: Can architects achieve aesthetically appealing buildings by using highly glazed facades while maintaining the visibility of the external glass surface in a hot arid climate by employing thermally insulated glass and internal curtains? To what extent do these interventions effectively shield the indoor environment from excessive heat gain?

The research objectives are threefold:

- To assess the thermal performance of the thermally insulated glazed facades in the buildings of Najran University, located in the hot arid climate of Saudi Arabia;
- To evaluate the effectiveness of interior roller blinds in reducing solar heat gain in highly glazed spaces in a hot arid climate;
- To explore architects' perceptions of glazing in building facades.

## 2. Materials and Methods

Recognising the divide between engineering's detailed performance analysis and architecture's artistic elements, the author utilised a mixed-methods approach to thoroughly investigate glazing in hot arid climates. Field tests yielded specific thermal measurements, complemented by architects' surveys that shed light on the artistic and experiential aspects

of glazing. This fusion of quantitative and qualitative approaches provides a broader perspective on glazing's effect on building design and the experiences of industry professionals.

### 2.1. Local Climate

Najran, located in southwestern Saudi Arabia, is positioned at the geographic coordinates of 17.62° north and 44.42° east. Figure 2 shows that under the Saudi Building Code (SBC-601), Najran falls within Zone 1, defined as a hot arid area [34]. This classification aligns with ASHRAE's climate zone 1B, which is categorised as hot and dry.

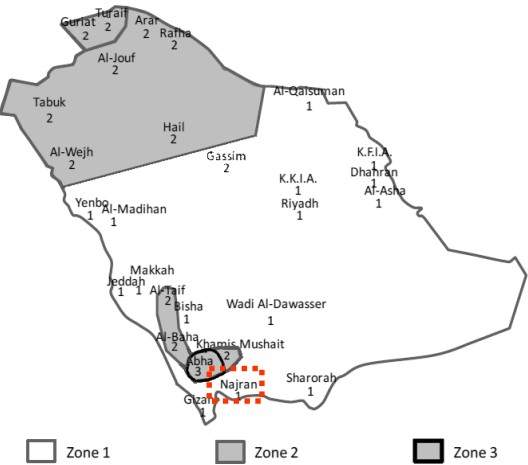

**Figure 2.** Classification of Najran as Zone 1 (hot arid) under SBC-601 [34].

Figure 3 offers a comprehensive overview of the fluctuations in solar radiation and air temperature throughout the year in Najran, revealing a distinct seasonal pattern. Both direct normal irradiance (DNI) and global horizontal irradiance (GHI) show a gradual increase from January, reaching their peaks around June, with GHI reaching approximately 600 W/m² and then decreasing towards December. This solar radiation trend closely corresponds to the variation in air temperature, which also rises to its highest levels during the summer, with maximum temperatures surpassing 40 °C or even higher. In contrast, the average low temperatures in winter hover around 15 °C. The annual average temperature is approximately 29 °C, with only minor variations. A distinctive characteristic of Najran's desert climate is the pronounced disparity between daytime and nighttime temperatures, reflecting the typical thermal dynamics of desert environments.

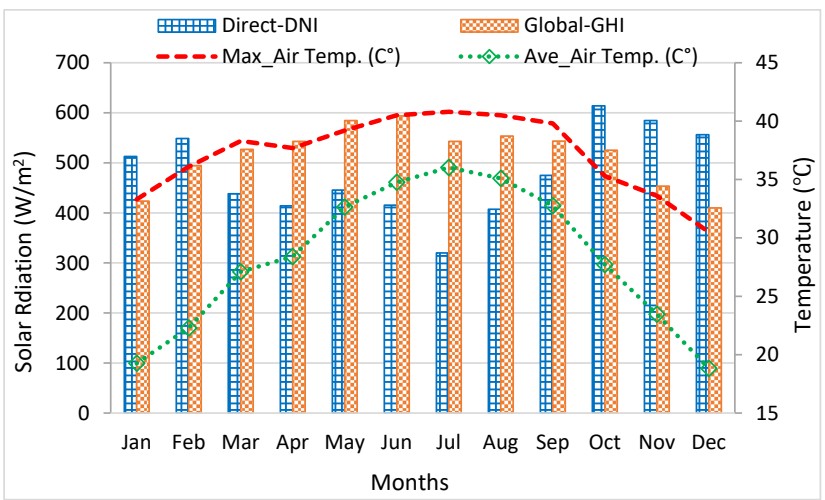

**Figure 3.** Monthly and yearly fluctuations in solar irradiance and air temperature. (Source: author, derived from NU weather station data, 2013–2016).

### 2.2. Case Study Definition and Instrumentation

The College of Engineering building at Najran University, presented in Figure 4—which provides a view of the building's layout and facade—is a three-storey structure with external dimensions of 202 m × 132 m. Its architectural design includes several fully glazed spaces, such as lecture halls, lounges, corridors and atriums. The specific highly glazed space under investigation is situated on the third floor of this building. The building features a roofing system comprising multiple layers, including a gravel stone layer, thermal insulation consisting of polystyrene, lightweight concrete and a concrete slab.

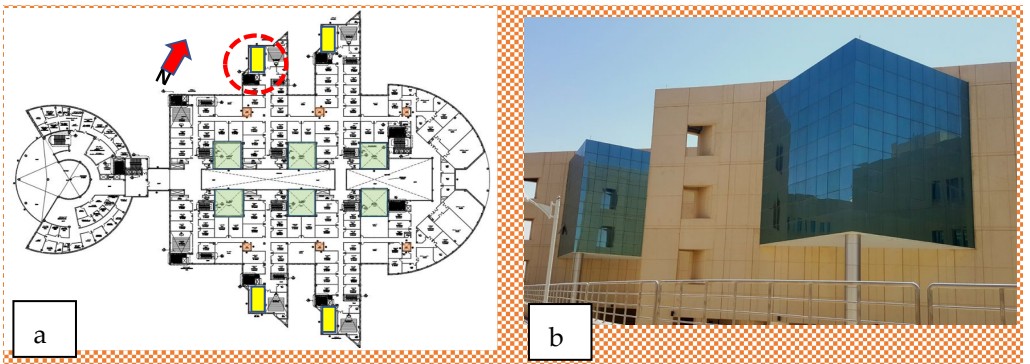

**Figure 4.** Visualisation of the measured lecture hall. (**a**) Plan of the third floor of the College of Engineering highlighting the lecture hall, (**b**) west-facade view of the lecture hall.

The lecture hall's external walls feature double-glazed facades, where each unit is composed of an outer pane made from 6 mm blue–green tinted glass with a low-E coating. This outer pane is paired with a 6 mm clear glass inner pane, creating a 12 mm air cavity between them. Figure 5 illustrates a cross section of the glazing system employed in the fully glazed spaces in Najran University buildings. The following is the theoretical basis for this glazing configuration:

(a) Double low-E tinted glazing consists of two glass layers separated by an air gap and hermetically sealed along their perimeter. The thicknesses are as follows: 6 mm tinted glass with low-E coating facing the cavity, a 12 mm air gap and 6 mm clear glass;

(b) The blue–green heavy tint of the glass further diminishes glare and solar heat gain by absorbing and reflecting a portion of the incoming solar energy;

(c) Interior roller blinds, which are constructed from light-grey PVC opaque material, regulate the amount of light and heat entering the space;

(d) The glazed facades are designed to be exposed to the outside view without any external shading. This design choice is made to ensure the facades remain visually appealing.

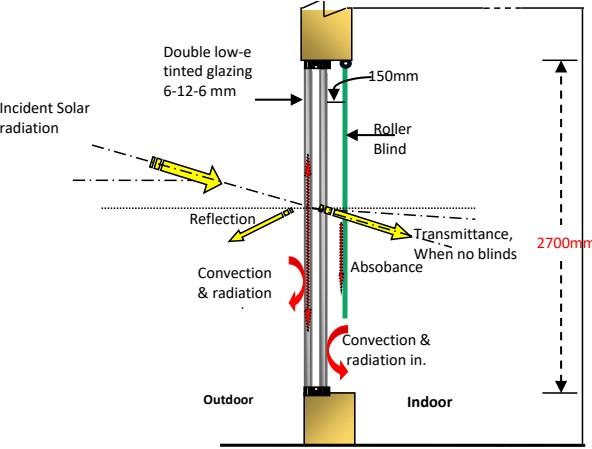

**Figure 5.** Schematic of glazing configuration and solar radiation interaction.

This configuration serves a crucial function in mitigating the effects of solar radiation, significantly reducing heat transfer from the external glazing surfaces to the interior. This reduction is achieved through a combination of mechanisms, including the reflection of a portion of the solar radiation, absorption of energy within the tinted glass and roller blinds, and the restriction of convection and radiation heat transfer through the air gap and low-E coatings. Table 1 provides information on the optical properties of the glazing material supplied by Saudi American Glass (SAG), with data sourced from the General Management of Projects, Maintenance, and Facilities at Najran University.

**Table 1.** Optical characteristics of SAG glazing installed in the College of Engineering building (Source: NU, General management of projects, maintenance and facilities).

| Description | | Air Filled | VL % | | | SHGC % | U-Value Summer W/m²K |
|---|---|---|---|---|---|---|---|
| Coating | Substrate | | Tv | Reflectance | | | |
| | | | | Out | In | | |
| SS-08- | Blue Green | 6 mm/12 mm/6 mm | 6 | 32 | 49 | 0.15 | 2.42 |

Data collection for this study was conducted from 1 to 18 June 2023, focusing on a lecture hall strategically positioned to receive direct solar radiation from both the southwest and northwest directions. Detailed indoor environmental data were gathered using three LSI-R-Log data loggers equipped with eight sensors. Outdoor environmental conditions were monitored through a rooftop weather station at the college. These instruments simultaneously recorded various parameters, including indoor and outdoor air temperatures, inner surface glazing temperature, air velocity, heat flux and horizontal global solar radiation (Figure 6 and Table 2). The sensors were positioned at the room's centre, approximately 1.0 m above the floor, and mounted on two tripods. Data loggers were programmed to continuously record readings at 5 and 10 min intervals over a 24 h period. Manual temperature readings were periodically taken using both a standard thermometer and an IR thermometer to enhance data accuracy. This study explores different roller blind configurations, including those fully closed, partially open and fully open (Figure 7).

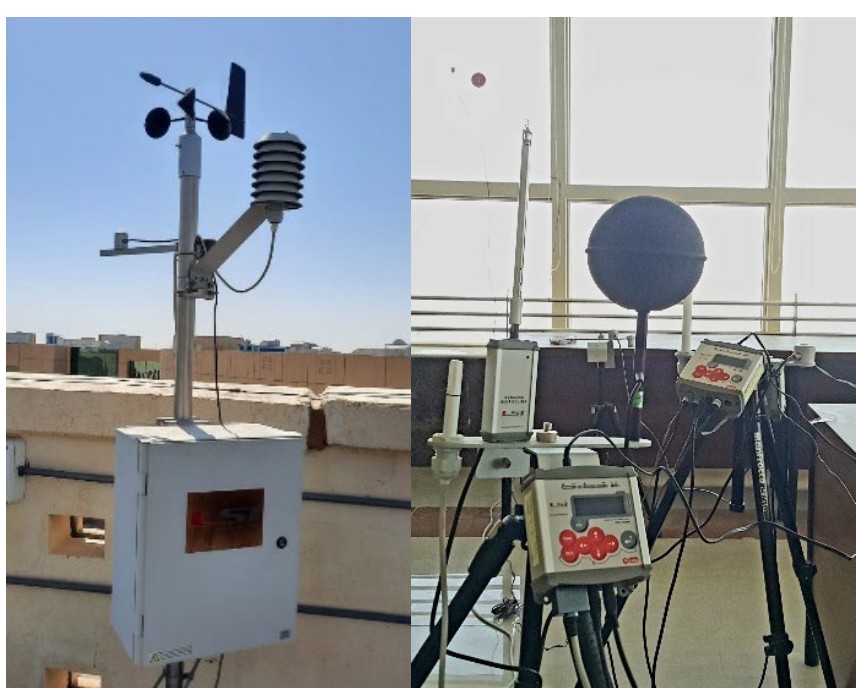

**Figure 6.** Data loggers for indoor and outdoor environments (LSI-R-Log and LSI-E-Log Weather Station).

**Table 2.** Technical specifications of the sensors used in the investigation.

| Instruments | Measuring Range | | Resolution |
|---|---|---|---|
| Three LSI-R-Log, data loggers | Surface temperature | −40 °C to +80 °C | ±0.01 °C |
| | Air temperature | −40 °C to +80 °C | ±0.01 °C |
| | Glob temperature | −40 °C to +80 °C | 0.01 °C |
| | Heat flux | $-2000 \div +2000$ W/m$^2$ | 50 µV/W/m$^{-2}$ |
| | Air speed | $0.01 \div 20$ m/s | 0.01 m/s |
| | Lux | Human eye response (CIE) | 3%, Uncertainty |
| LSI-E-Log, data logger—outdoor | Air temperature | $-50 \div 70$ °C | 0.1 °C (@0 °C) |
| | Air speed and direction | 0 to 75 m/s | 0.07 m/s |
| | Pyranometer | 0 to 2000 W/m$^2$ | $10 \div 15$ µV/W/m$^2$ |

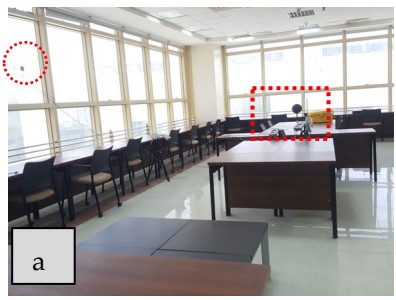 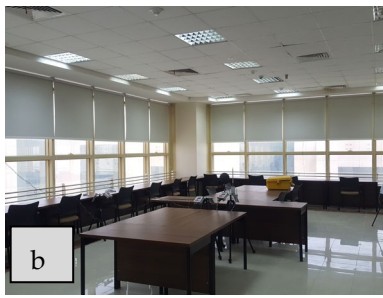 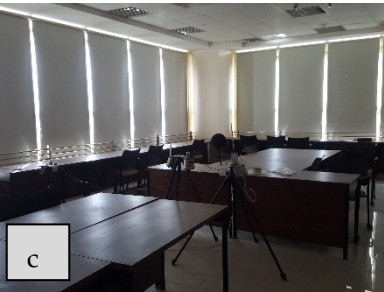

**Figure 7.** Arrangement for indoor measurements: (**a**) uncovered glazing measurement, (**b**) 50% closure of roller blinds, (**c**) full closure of roller blinds.

*2.3. Architects' Perceptions of Glazing in Building Facades*

This section delves into the insights of architects who are staff or alumni of Najran University who have direct experience of the university's buildings. This study examines their professional assessments regarding the aesthetic appeal, functional and climatic suitability of glazed facades in buildings in hot arid climates. An online survey provided a structured quantitative framework for these architects to rate the importance of glazing attributes on a scale from 1 (least important) to 5 (most important). The gathered data, presented in percentages, offer an aggregate perspective on the value placed on each aspect of glazing.

**3. Results and Discussion**

In this section, results and discussions are presented, detailing experimental conditions and evaluating the thermal performance of glazing materials, as well as the impact of interior roller blind configurations on the indoor environment. Additionally, the effects of these factors on lighting quality and environmental outcomes are explored. Finally, architects' perspectives on the aesthetic, functional, and environmental impacts of glazing in hot arid climates are examined, emphasizing the balance between aesthetic appeal and thermal efficiency in architectural designs.

*3.1. Schedule of Measurements*

The data collection period extended over 18 days, with each case corresponding to different ratios of manually controlled roller blinds (0%, 50% and 100% closed) observed for 6 days each. Figure 8 illustrates the average hourly values recorded for the whole days of various outdoor parameters, including outdoor air temperatures, air speed and global solar radiation associated with indoor air temperature, recorded from 1 to 18 June 2023. The data show variations in outdoor air temperature over the period. These fluctuations will be accounted for when assessing the thermal performance of the space under varying

roller blind configurations. For precise analysis, the study concentrates on specific days with similar outdoor conditions: 10 and 11 June for the 100% closed blinds scenario, 14 to 17 June for the 50% closure and 6 to 9 June for the scenario without blinds (0% closed).

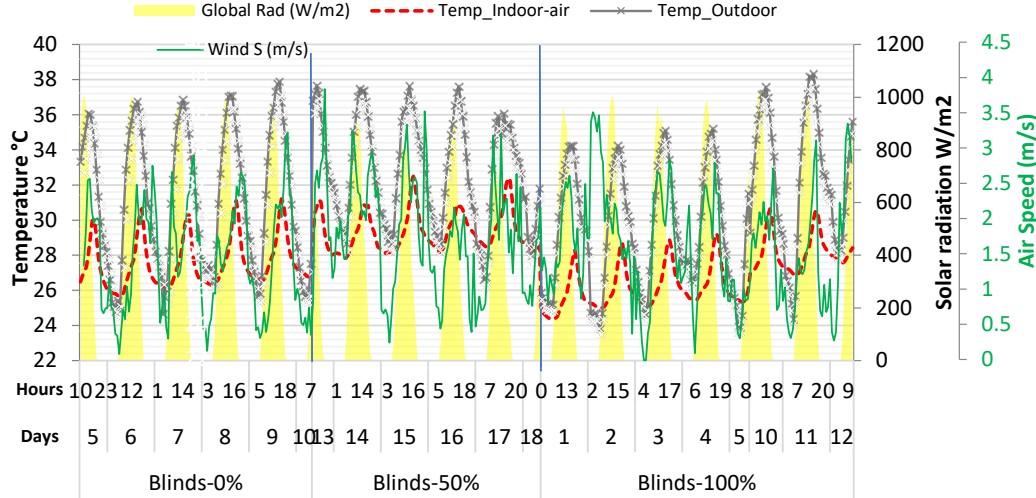

**Figure 8.** Schedule of measurements of all cases from 1 to 18 June 2023. Temperature against solar radiation and outdoor air speed is given.

### 3.2. Glazing's Thermal Performance

In unoccupied spaces where air conditioning is not in operation, as in the investigated lecture hall, the primary consideration should be glazing performance rather than indoor thermal comfort. This section of the assessment evaluates the efficiency of the selected glazing in Najran University buildings, specifically in managing heat gain, focusing on the measured solar heat gain, which is essentially the heat added to a space by the solar radiation transmitted through the glazing and absorbed by various surfaces.

The data in Figure 9 present two different measurements over a 24 h period, illustrating the relationship between the temperature differential across the glazing (ΔT) and the corresponding heat flux, which offers insights into the glazing's capability to reduce solar heat gain. The ΔT line fluctuates over time, showing positive values for most of the day and reaching a peak at approximately 9.2 °C, suggesting that the glazing helps maintain a lower indoor temperature compared with the outdoors. In contrast, the heat flux values are negative throughout, indicating an inward conduction of heat. The most pronounced inward heat flux, at −105 W/m$^2$, coincides with the maximum ΔT, reflecting the substantial heat absorption by the heavily tinted glazing. This absorption is in line with the glazing's low SHGC of 0.15, which is significantly lower than that of standard commercial double glazing with SHGC values between 0.31 and 0.84 [35]. While the low SHGC suggests reduced solar heat transmission, the notable inward heat flux, particularly during periods of intense solar radiation, points to considerable heat transfer into the building. This data is crucial for understanding the thermal performance of the glazing on the facade of the College of Engineering building. However, the figure demonstrates a time shift between ΔT and heat flux values, attributed to a pronounced increase in inward heat flux when the glazed facade faces solar radiation around 3 pm. This condition leads to a surge in indoor temperature, which, in turn, reduces the ΔT, illustrating the dynamic interaction between heat flux and indoor thermal conditions. In a hot arid climate, buildings often use thermal mass to help stabilise indoor temperatures. The results suggest that the building has a good thermal mass that helps keep the indoor temperature lower than the outdoors even with significant heat flux through the glazing.

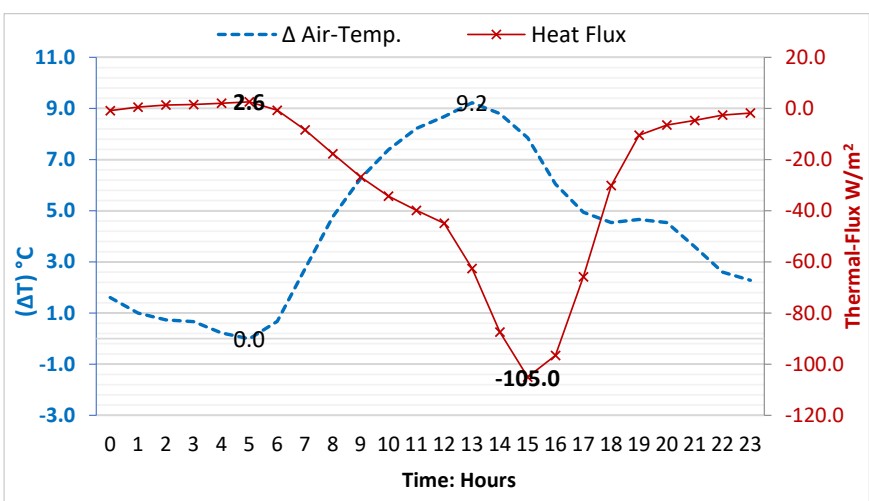

**Figure 9.** Steady state of solar heat gain, the heat added to a space by the sun's radiation measured by (T2-T1) along with heat flux through glazing.

The relationship between solar radiation and the indoor thermal profile in the lecture hall is depicted in Figure 10. As solar irradiance increases, reaching its peak at approximately 1000 W/m$^2$ at noon, the outdoor temperature, marked by the red dashed line, rises accordingly, which is expected because of the direct solar heating of outdoor surfaces and air. Indoor air temperature rises from roughly 26 °C at 6 am to 30.5 °C at 4 pm, when the outdoor air temperature is 36.2 °C. The MRT, inferred from the globe temperature, exhibits more variability, ascending from 26 °C at 6 am to a peak of 32 °C at 4 pm, which suggests a radiant heat impact on the space. This effect is further accentuated by the inner glass temperature, which rises to approximately 43 °C at 4 pm when direct solar radiation strikes the west facade of the investigated lecture hall. This temperature is notably higher than both the indoor air and globe temperatures. This peak indicates substantial solar heat absorption by the glazing, contributing to the radiant heat within the lecture hall. The glazing, which absorbs solar radiation, becomes a radiative heater itself, elevating the MRT and, by extension, the perceived temperature by occupants. The radiant temperature difference, at its peak, is around 1.5 °C between the globe and indoor air temperatures, highlighting the net radiant heat exchange influencing the indoor thermal profile. This effect would need to be considered in air-conditioning operations to ensure students' comfort, as the actual air temperature may not fully represent the thermal sensation experienced due to radiant heat exchange.

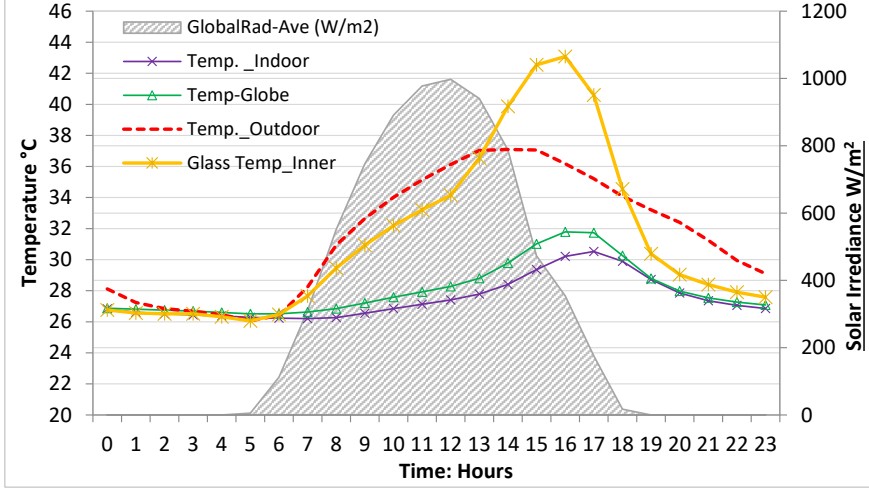

**Figure 10.** Correlation between indoor thermal profiles and external environmental conditions.

Figure 10 reveals a discernible temporal disconnection between indoor and outdoor temperature fluctuations, underscoring the building's thermal management capabilities. At 1 pm, the peak of outdoor temperature is 37 °C, whereas by 5 pm, the indoor temperature ascends to a moderate 30.5 °C despite a decline in the outdoor temperature to 35.2 °C. The observed thermal delay in the indoor environment can be attributed to the synergistic effects of direct solar radiation on the glazing when the sun shifts towards the western facade, coupled with the building's thermal inertia. This inertia enables the structure to function as a thermal buffer, gradually absorbing heat and then slowly releasing it over time. This finding indicates that the glazing strategically mitigates heat transfer, effectively decoupling the indoor climate from external temperature spikes, thus enhancing occupant comfort and reducing the demand on cooling systems.

### 3.3. Interior Shading Effectiveness

As previously mentioned, the glazing in the examined space demonstrates a notably low SHGC of 0.15, effectively impeding 85% of the solar radiation from penetrating the interior space. This high level of solar control is achieved through a combination of reflection, absorption within the glazing itself and the transmission of long-wave IR radiation, which contributes significantly to the reduction of solar heat gain. The addition of interior blinds offers an extra layer of thermal regulation, markedly influencing the heat transfer dynamics by modifying convective heat exchange at the glazing's inner surface. The impact of the blinds on the building's thermal performance is quantitatively assessed by calculating the HTC, a measure that is instrumental in evaluating the efficacy of temperature regulation within the space. A lower HTC signifies superior insulative properties, correlating with enhanced energy conservation and occupant comfort.

$$h = \frac{q}{\Delta T} \, (\mathrm{W/m^2/K}) \tag{2}$$

where q is the heat flux ($\mathrm{W/m^2}$) and $\Delta T$ is the difference in temperature between the glazing surface and air within the space (K). Equation (2) is commonly referred to as Newton's law of cooling [36].

This HTC has been meticulously computed and is illustrated in Figure 11, showcasing the thermal profiles for three specific scenarios: the absence of blinds, 50% closed blinds and completely closed blinds. These scenarios present a comprehensive view of how varying levels of shading contribute to the interior thermal environment, ultimately guiding strategic decisions for passive cooling and energy efficiency enhancements in building design. As depicted in the figure, the HTC is reduced when the blinds are either partially or fully open, demonstrating the blinds' role in impeding heat ingress, primarily by reducing convective heat transfer and curtailing the influx of solar radiation. Notably, the HTC does not remain static, even with the blinds' position held constant, underscoring the influence of external variables such as fluctuations in outdoor temperature or the solar trajectory. This finding indicates that blinds serve as an adjustable barrier to heat transfer. A lower HTC reflects better insulation properties, leading to improved energy efficiency and enhanced comfort for occupants. The findings are summarised as follows:

- When the blinds are completely open (0%), the HTC values start at around 6 $\mathrm{W/m^2K}$ in the early morning and reach peaks close to 8 $\mathrm{W/m^2K}$, coinciding with the times of elevated outdoor temperatures. The average daytime HTC in this scenario is 7.1 $\mathrm{W/m^2K}$;
- When the blinds are 50% closed, the HTC exhibits a noticeable reduction, peaking at approximately 5.6 $\mathrm{W/m^2K}$. This reduction suggests a moderate decrease in heat transfer, likely due to the blinds' partial shading effect. The average daytime HTC here is 5 $\mathrm{W/m^2K}$;
- In the scenario where the blinds are fully closed (100%), despite higher outdoor air temperatures compared with that in other scenarios, the HTC drops significantly, thereby indicating the blinds' efficiency in insulating the interior from external heat.

In this case, the HTC values typically range from 3 W/m$^2$K to 5 W/m$^2$K, indicating the most effective insulation from the blinds. The average daytime HTC in this setting is 4.2 W/m$^2$K.

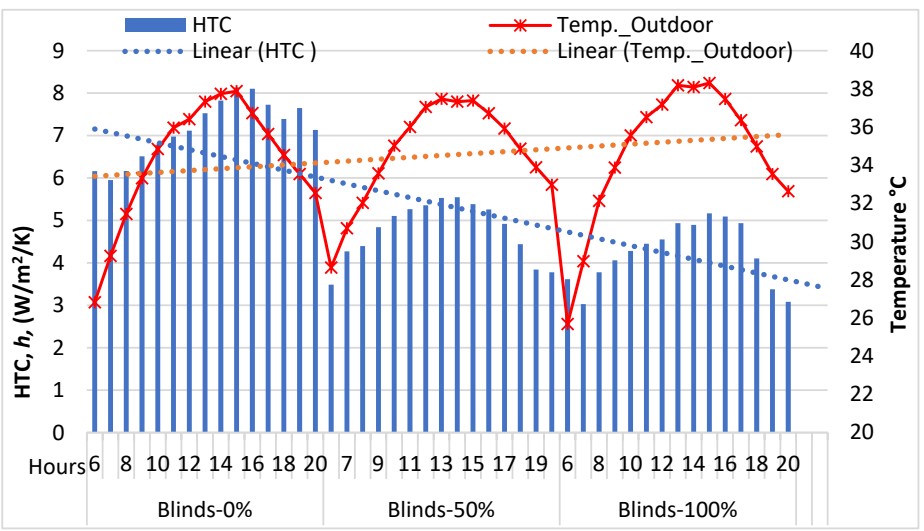

**Figure 11.** Impact of interior roller blinds on HTC and correlation with diurnal outdoor temperature variations.

Figure 12 corroborates the observation that blinds play a role in modulating indoor thermal dynamics. With the blinds fully retracted (0%), the inner surface of the glass registers a temperature of 43.1 °C at 3 pm, coupled with an elevated air speed when compared with other scenarios, which aligns with the increased HTC values. The temperature gradient between the hot glass surface and the relatively cooler air farther away causes air movement. Closing the blinds by 50% results in a noticeable dip in the glass surface temperature, peaking at 42 °C, and the HTC, reaching 5.6 W/m$^2$K. Notably, when the blinds are completely closed (100%), the glass surface temperature rises unexpectedly, peaking at 47.8 °C at 4 pm. This finding suggests that while the blinds mitigate direct solar radiation, the air confined between the blinds and the glass may amplify the glass temperature. Despite this condition, the HTC declines significantly, which underscores the blinds' effectiveness in insulating the space from external thermal variations.

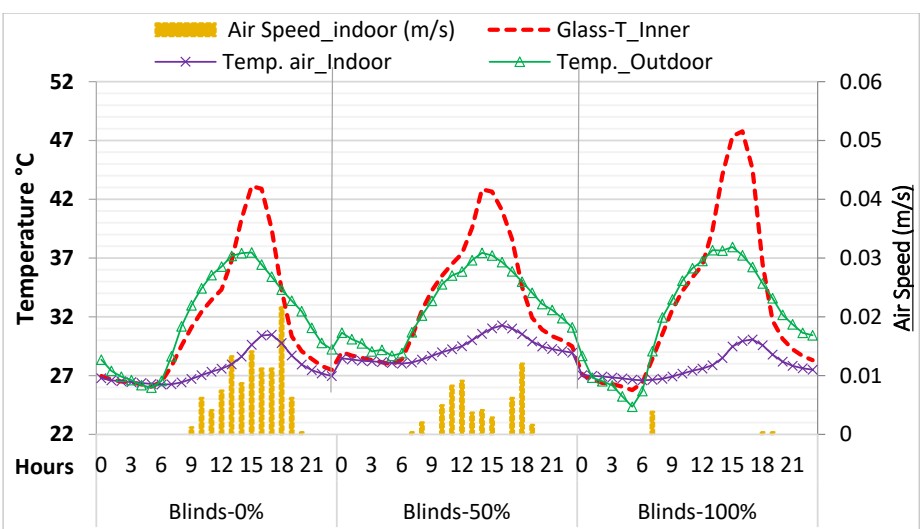

**Figure 12.** Impact of blinds on glass surface temperatures and indoor heat transfer dynamics.

### 3.4. Further Discussion on Lighting and Environmental Impacts

Previous studies [37,38] explored daylight illumination in the space under investigation. However, Figure 13, depicting lux measurements in the centre of the lecture hall, reveals the impact of heavily tinted, low-E coated double-glazing on light transmission and indoor illumination. On days with the blinds fully open, the lux levels reach up to 150 lux, a moderate intensity due to the tinting and low-E coating, which manage solar radiation and reduce potential glare. Interestingly, the indoor air temperature peaks are aligned with the lux peaks, suggesting that despite the tinting and low-E coating, a noticeable heat gain that affects the indoor climate remains. When the blinds are drawn to 50%, the lux levels decrease significantly, indicating less light penetration and a corresponding moderate reduction in the indoor air temperature peaks, which illustrates the combined thermal control properties of the blinds and the glazing. With the blinds fully closed, the lux measurements drop below 50 lux, which contributes to a further stabilisation of the indoor air temperature, underscoring the synergy between the glazing's properties and the blinds in regulating both light and heat, which is essential for maintaining a comfortable indoor environment.

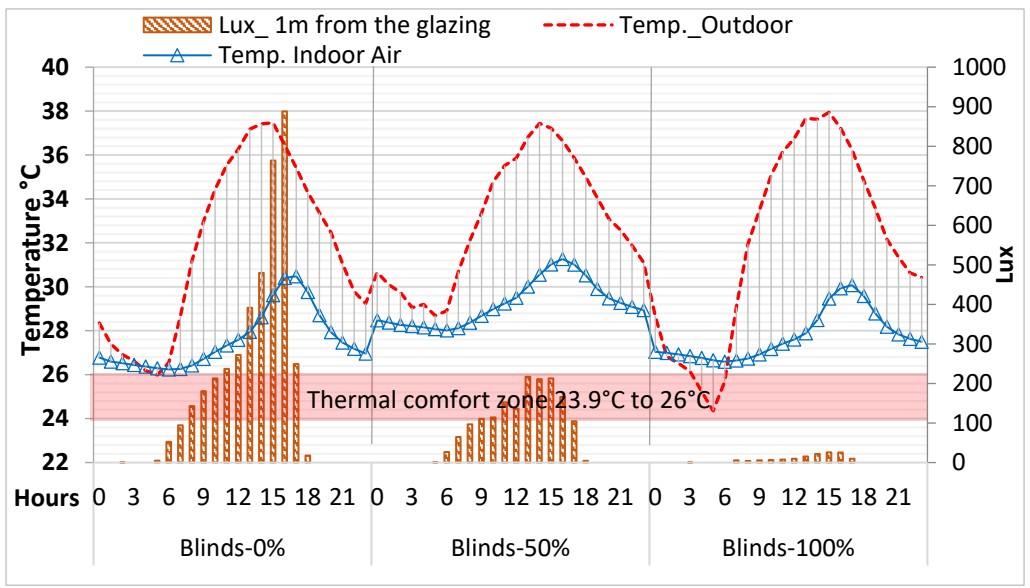

**Figure 13.** Light and temperature control using blinds, assessing VT and solar heat gain.

In redesigning the lecture hall for optimal performance, a key focus should be on improving the light-to-solar gain (LSG) ratio. This ratio measures the ability of glazing materials to offer adequate VL while reducing solar heat gain [16]. The current glazing of the investigated lecture hall, characterised by a low SHGC, suggests the potential improvement of VL transmittance (VT). Selecting glazing with a higher VT but still retaining a low SHGC is crucial to enhance the LSG ratio. Furthermore, incorporating dynamic shading systems can provide tailored control over light and heat, adapting to changes in solar exposure. Implementing advanced insulated glazing with dynamic smart films, such as PDLC film, could dynamically respond to varying solar radiation intensities, thereby efficiently distributing daylight in the absence of direct solar radiation and simultaneously reducing direct solar gain.

Overall, the glazing installed in the building significantly limits solar radiation entry, evidenced by the up to 9.2 °C temperature difference between the indoors and the outdoors. Specifically, Figure 13 highlights that without blinds, indoor temperatures soar to 30.4 °C by 5 pm, exceeding the Saudi Building Code's (SBC) comfort ceiling of 26 °C. This scenario indicates that while the current design curtails some cooling needs, there is potential for further enhancements. Integrating daylighting optimisation and passive design principles in line with SBC standards is vital to achieve a more effective balance between maintaining

comfortable temperatures and conserving energy. Such improvements will not only enhance thermal comfort but also contribute to environmental protection by lowering $CO_2$ emissions, advancing sustainability efforts at Najran University.

### 3.5. Architects' Perceptions of Glazing in Najran University Buildings

This section analyses the perspectives of 35 architects regarding the aesthetic, functional and environmental implications of glazing in building designs at Najran University. Figure 14 shows a clear preference for functionality among the architects, with a striking 71% of responses indicating that it is the most important criterion (level 5). This strong emphasis on functionality likely stems from the architects' recognition of the critical role of glazing in the practical aspects of a building's performance, such as natural lighting, insulation and overall environmental interaction. Aesthetic appeal is the second most valued criterion, with more than half of the respondents (52%) giving it the highest level of importance. This finding suggests that architects consider the visual impact of glazing to be nearly as crucial as its functional role. The importance of aesthetics highlights the architects' focus on the design and appearance of building facades, which contribute significantly to the overall architectural expression and the way a building is perceived. Energy efficiency also receives considerable attention, with 44% of responses ranking it the most important criterion. Its close rating to aesthetic appeal suggests that energy-efficient design is nearly as prioritised as the visual aspects, reflecting an integrated approach to aesthetic and sustainable design practices. Cost-effectiveness has a relatively balanced spread across levels 3, 4 and 5, indicating a moderate importance.

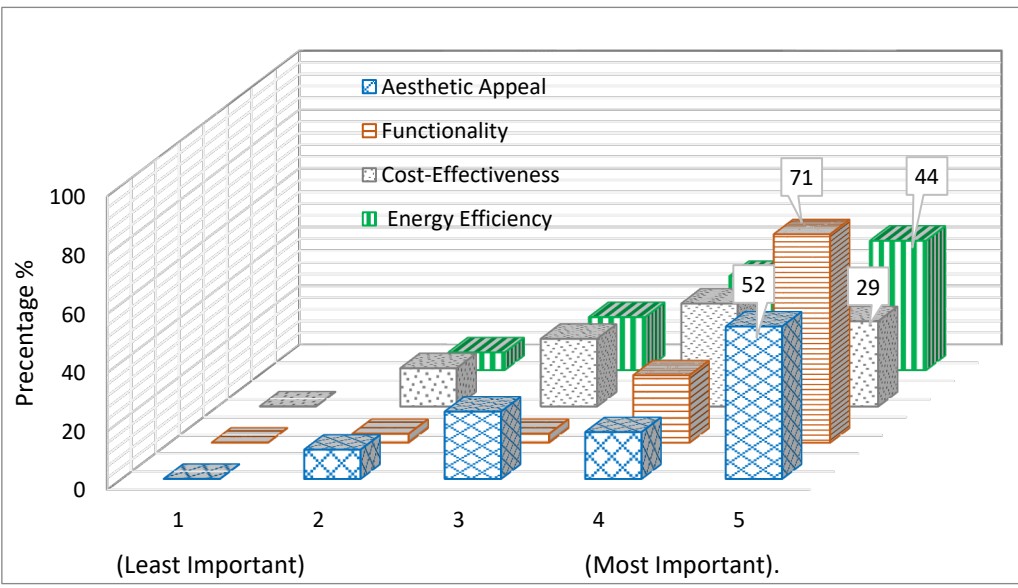

**Figure 14.** Architects' prioritisation of criteria in glazed building facades at Najran University.

Figure 15 provides insights into the perceptions of the effectiveness of glazing in addressing specific concerns related to building facades at Najran University. The data are categorised into three distinct queries. The first query, assessing the effectiveness of Najran University's glazed facades in addressing the hot arid climate, shows a notable distribution across the scale, but with the highest concentration of responses in the mid to high importance range, suggesting that many respondents see glazing as fairly effective in mitigating harsh climatic conditions. The second query evaluates the impact of glazed facades on indoor comfort levels. A significant majority (73%) rated this as the most important. This overwhelming response indicates a strong consensus among architects that glazed facades play a crucial role in influencing the comfort levels within the buildings, likely due to their ability to control light and heat transmission. The third and final query concerns the importance of thermal insulation in glass facades. It is considered by 61%

of the respondents to be the most important, marking it at level 5. This high percentage underscores a widespread recognition of the critical role of thermal insulation in enhancing the performance of glazed facades. Concerning the effectiveness of Najran University's glazed facades in addressing the hot arid climate, only 24% of respondents ranked it as having the highest level of effectiveness. This finding signifies a critical viewpoint, indicating that although the glazed facades, as evidenced in the earlier thermal discussion, may be operating adequately, their overall effectiveness in such a challenging climate is not entirely satisfactory, demanding further attention.

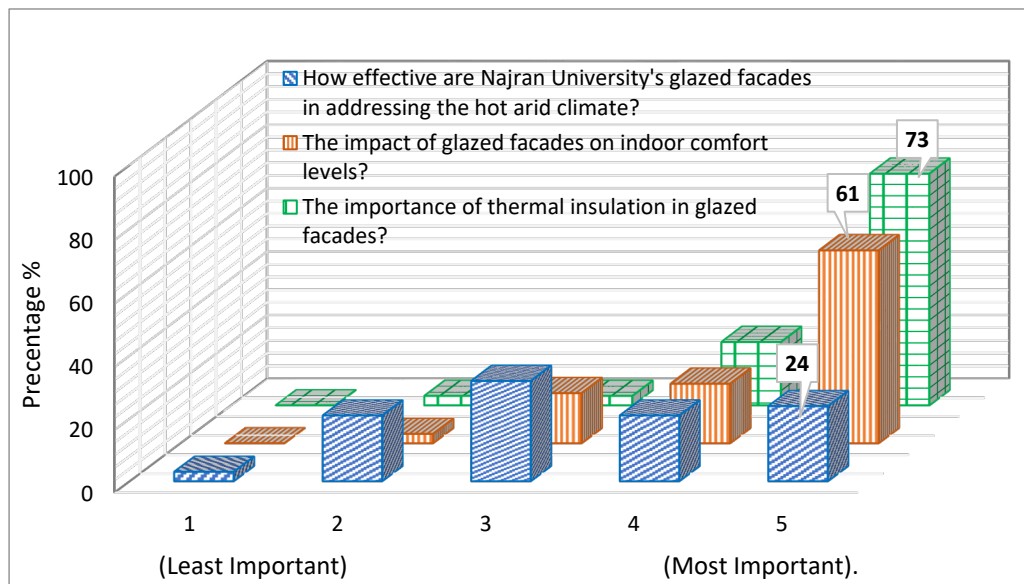

**Figure 15.** Evaluating the role of glazed facades in climate adaptation at Najran University.

Overall, the results suggest that while aesthetics are valued, the practical aspects of functionality are of paramount concern. Moreover, architects recognise the crucial influence of glazed facades on the comfort levels in buildings and the essential role of thermal insulation in enhancing facade efficacy. This finding could reflect a trend in architectural design, prioritising sustainable and efficient building practices without compromising on the visual appeal of the buildings.

### 3.6. Limitations and Uncertainties

With measurements taken over a period of 18 days, variations in outdoor weather conditions can introduce variability into the data. This variability makes it challenging to establish a controlled baseline for comparison. A pretest-posttest design that does not account for identical outdoor weather conditions can yield skewed results, affecting the reliability of the conclusions about the blinds' effectiveness and the glazing's thermal performance. A careful selection of days for comparison is essential to mitigate this limitation. Days with similar weather conditions were chosen to ensure that the data reflect the performance of the glazing and blinds rather than the fluctuations in the external environment. This approach allows for a more accurate assessment of the glazing's properties and the impact of the internal blinds on indoor temperature conditions, leading to more reliable conclusions.

The air in both the investigated space and the adjoining corridor exhibits temperature disparities, leading to variations in density. Gaps around the door allow for uncontrolled air leakage, contributing to this phenomenon. The warmer air in the lecture hall generates a minor pressure differential close to the door, facilitating air movement and leading to a marginal heat loss from the lecture hall. This thermal infiltration around the door can lead to a decrease in the indoor air temperature of the non-conditioned space being studied, altering the temperature differential. This change affects the performance metrics and outcomes of the investigated facade system. Therefore, accounting for the impact of

infiltration towards adjacent spaces is important when assessing the glazed facade's ability to control heat gain in future studies. Furthermore, the influence of adjacent air-conditioned areas could skew the results of the effects of the glazing and interior roller blinds in the investigated space. While acknowledging the limitation of not measuring the heat flux between adjacent spaces and the lecture hall, a crucial detail to highlight is that the partition walls are 20 cm thick concrete walls, and one of these two walls is adjacent to bathrooms equipped with mechanical ventilation, which reduces thermal exchange through these boundaries. The presence of substantial physical barriers and a consistently closed door significantly reduces the thermal exchange. This condition is a limitation due to the lack of sensors and could be covered in future studies.

## 4. Conclusions

This study conducts field measurements to assess the thermal performance of aesthetically driven glazed facades in a hot arid climate, exposing an architectural paradox that lies in the conflict between achieving aesthetically pleasing glazed facades and the need for energy efficiency in such climates. The findings can help architects and engineers in enhancing building aesthetics and sustainability while prioritising occupant well-being, aligning with Najran University's vision for a sustainable campus environment.

The following are the key conclusions from the on-site thermal measurements:

- The existing glazing, with a low SHGC of 0.15, effectively reduces solar radiation transmission, as indicated by lower indoor temperatures compared with the outdoors, with a peak temperature difference of about 9.2 °C;
- The glazing significantly absorbs solar heat, raising the inner glass temperature to 43 °C and acting as a radiative heater. As a result, the MRT is increased by 1.5 °C compared with indoor air temperatures, suggesting an impact on the indoor thermal profile despite the glazing's effectiveness;
- The adjustable blinds have a notable impact on the HTC, effectively regulating temperature by obstructing the natural convection currents, which are usually affected by the temperature of the glass surface;
- With blinds fully open (0%), the HTC averages 7.1 W/m$^2$K during the day. When the blinds are closed 50%, the HTC decreases to a 5 W/m$^2$K daytime average. Fully closed blinds (100%) significantly reduce HTC to an average of 4.2 W/m$^2$K during the day, demonstrating their efficiency in thermal insulation;
- Despite the glazing's heavy tint and blind usage, a correlation between indoor light levels and temperatures is observed, indicating daylight's contribution to thermal load. This condition highlights the need to optimise the glazing's VT to improve the LSG ratio. Employing real-time responsive shading systems and daylighting simulation tools for predictive modelling could further enhance environmental control in spaces.

The study's survey reveals architects' preference for glazed facades, with a majority prioritising functionality (71%) and aesthetics (52%). In contrast to the emphasis on visuals, challenges such as energy efficiency (44%) are delicately weighed. Notably, 73% of architects underscore the significance of thermal insulation associated with glass facades, highlighting the need for a comprehensive approach to address both visual and thermal aspects. Regarding Najran University's glazed facades and their effectiveness in the hot arid climate, only 24% of respondents rated them as highly effective. This percentage suggests a critical perspective, indicating that the overall effectiveness of glazed facades in this challenging climate demands further attention despite their satisfactory performance in specific aspects exhibited in the thermal performance results.

Future studies can explore different related directions. The use of adaptive shading systems, such as automated blinds or dynamic glass, is recommended to respond effectively to varying outdoor conditions, thus balancing heat gain control and daylight utilisation. An integrated design approach, considering both aesthetics and functionality, is essential. This approach should involve collaboration among architects, engineers, and sustainability experts from the early stages of design, ensuring a harmonious blend of form and function.

**Funding:** This research was funded by Najran University, grant number NU/NRP/SERC/12/26. The APC was funded by the Deanship of Scientific Research at Najran University.

**Institutional Review Board Statement:** Not applicable.

**Informed Consent Statement:** Informed consent was obtained from all subjects involved in the study.

**Data Availability Statement:** The data presented in this study are available on request from the corresponding author.

**Acknowledgments:** The author would like to extend their gratitude to the Deanship of Scientific Research at Najran University for their financial support through the Research Priorities and Najran Research funding program grant code (NU/NRP/SERC/12/26).

**Conflicts of Interest:** The authors declare no conflicts of interest.

## Abbreviations

| | |
|---|---|
| DNI | Direct normal irradiance, $W/m^2$ |
| GHI | Global horizontal irradiance, $W/m^2$ |
| HTC, (h) | Heat transfer coefficient, $W/m^2K$ |
| IR | Infrared |
| Low-E | Low-emissivity |
| LSG | Light to solar gain, % |
| MRT | Mean radiant temperature, $°C$ |
| q | Heat flux ($W/m^2$) |
| SAG | Saudi American Glass |
| SHGC | Solar heat gain coefficient, % |
| UV | Ultraviolet |
| VL | Visible light |
| $\alpha_s$ | Solar absorptance |
| $\tau_s$ | Direct solar transmittance |

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
