# Peer review of "Aesthetic and Thermal Suitability of Highly Glazed Spaces with Interior Roller Blinds in Najran University Buildings, Saudi Arabia"

_sustainability, doi:10.3390/su16052030_

Round 1
Reviewer 1 Report
Comments and Suggestions for Authors
The manuscript “Aesthetic and thermal suitability of highly glazed spaces with interior roller blinds in Najran University buildings, Saudi Arabia” evaluates the effectiveness of Low-E double dark glass with internal roller blinds in reducing heat gain into glazed spaces in hot, arid climates. In addition, the opinions of architects on facades with this design are discussed.
The reviewer believes that the manuscript as a whole is interesting, but it contains a number of shortcomings and needs to be improved.
1. A significant amount of the manuscript is occupied by general discussions that will be of little interest to readers. In this case, individual thoughts are repeated several times (lines 485-490).
2. Aesthetic aspects are discussed with architects who are employees or graduates of Najran University, which may affect their objectivity. It is necessary to add the opinions of independent architects working in similar climatic conditions.
3. The description of Figure 4 (lines 235-236) does not correspond to its content.
4. Figure 8 and its description (lines 300-315) do not correspond to each other. Also in this figure the units of solar radiation are incorrectly indicated.
5. The author writes that the incoming heat flux, equal to -105 W/m², coincides with the maximum value of ΔT (lines 330-332). However, in Figure 9 there is a significant time shift between these values, which should be explained.
6. It is necessary to provide additional information on the space between the glasses: is it sealed? What is it filled with?
7. The author writes that two walls of the lecture hall are connected to air-conditioned rooms. It is worth assessing the heat flow through these walls because it can affect the air temperature.
Reviewer 2 Report
Comments and Suggestions for Authors
Dear author,
The study includes an important experimental section. After reviewing the manuscript, I suggest accepting it for publication after minor modifications:
Please add abbreviations table.
”Heat Transfer Coefficient” was defined more than once in the text.
Please delete line 565.
Please use air speed (as in Figure 12) and not wind speed (as in Figure 8).
Please add the unit to the last column of Table 1.
Please delete lines 188–194.
The aim of the study was repeated several times in the text. Please place it only at the end of the introduction.
Warm regards
Reviewer 3 Report
Comments and Suggestions for Authors
This manuscript examines the thermal performance of aesthetically pleasing glazed facades in a hot arid climate, highlighting the conflict between aesthetics and energy efficiency. The research provides insights for architects and engineers to enhance building aesthetics while prioritizing occupant wellbeing and sustainability. Key findings show that the existing glazing reduces solar radiation transmission with a low Solar Heat Gain Coefficient but significantly absorbs solar heat, raising the inner glass temperature and impacting indoor thermal conditions. Adjustable blinds regulate temperature and demonstrate efficiency in thermal insulation. However, there is a correlation between indoor light levels and temperatures, indicating the need for optimizing Visible Light Transmittance. Architects prefer glazed facades for functionality and aesthetics but weigh energy efficiency challenges. Future studies recommend adopting adaptive shading systems and an integrated design approach involving collaboration among architects, engineers, and sustainability experts from the early stages of design. I have some small concerns,
(1) Figure 3: The units on the left of Figure 3(W/m2) should use superscripts. The temperature ° C is incorrectly marked
(2) Line 264, Page 8: Whether the weather is taken into account in the selected days.
(3) Line 379, Page 12: The theoretical basis for this "infiltration" argument should be cited.
(4) Figure 11: The two dotted lines are not obvious.
(5) Limitations and Uncertainties: How to avoid these situations as much as possible, it is necessary to explain more.
(6) For better results, is it possible to consider using solar energy for storage applications (such as combining Windows with solar cells)? Combining aesthetics with thermal adaptability
Comments on the Quality of English Languagenone
Round 2
Reviewer 1 Report
Comments and Suggestions for Authors
The author considered all the reviewer's comments and made appropriate corrections to the text. The manuscript may be accepted.
Author Response
Thank you for your insightful feedback. Following your guidance, I've thoroughly revised the manuscript. This effort has significantly enhanced the manuscript's quality, aligning it more closely with the journal's standards. Your input has been invaluable in this process.